# Regulation and Role of Transcription Factors in Osteogenesis

**DOI:** 10.3390/ijms22115445

**Published:** 2021-05-21

**Authors:** Wilson Cheuk Wing Chan, Zhijia Tan, Michael Kai Tsun To, Danny Chan

**Affiliations:** 1School of Biomedical Sciences, The University of Hong Kong, Pokfulam, Hong Kong; cwilson@hku.hk; 2Department of Orthopaedics Surgery and Traumatology, The University of Hong Kong-Shenzhen Hospital (HKU-SZH), Shenzhen 518053, China; tanzj@hku-szh.org (Z.T.); mikektto@hku.hk (M.K.T.T.); 3Department of Orthopaedics Surgery and Traumatology, The University of Hong Kong, Pokfulam, Hong Kong

**Keywords:** bone, osteogenesis, transcription factor, osteoblast, osteoblast differentiation, epigenetics, microRNA, circadian rhythm, skeletogenesis

## Abstract

Bone is a dynamic tissue constantly responding to environmental changes such as nutritional and mechanical stress. Bone homeostasis in adult life is maintained through bone remodeling, a controlled and balanced process between bone-resorbing osteoclasts and bone-forming osteoblasts. Osteoblasts secrete matrix, with some being buried within the newly formed bone, and differentiate to osteocytes. During embryogenesis, bones are formed through intramembraneous or endochondral ossification. The former involves a direct differentiation of mesenchymal progenitor to osteoblasts, and the latter is through a cartilage template that is subsequently converted to bone. Advances in lineage tracing, cell sorting, and single-cell transcriptome studies have enabled new discoveries of gene regulation, and new populations of skeletal stem cells in multiple niches, including the cartilage growth plate, chondro-osseous junction, bone, and bone marrow, in embryonic development and postnatal life. Osteoblast differentiation is regulated by a master transcription factor RUNX2 and other factors such as OSX/SP7 and ATF4. Developmental and environmental cues affect the transcriptional activities of osteoblasts from lineage commitment to differentiation at multiple levels, fine-tuned with the involvement of co-factors, microRNAs, epigenetics, systemic factors, circadian rhythm, and the microenvironments. In this review, we will discuss these topics in relation to transcriptional controls in osteogenesis.

## 1. An Exciting Era of Bone Biology

Our skeleton does not only provide mechanical support to facilitate locomotion and protect our internal organs. It is also a major reservoir of calcium and phosphate in our body. Bone also provides the marrow space as a niche for hematopoiesis. In the past decade, our knowledge of bone biology has expanded with advances in molecular cell technologies. State-of-the-art flow cytometry enabled enrichment of specific cell types, while high-resolution and real-time imaging tools facilitated the visualization of cells and interaction with their environment. Advances in genome editing have accelerated the generation of genetically modified cells and animal models, for functional tests and studies of bone diseases. Single-cell transcriptomic and *in vivo* cell lineage technologies have transformed our ability to identify novel transcription factor pathway interactions in osteogenesis. New cell types such as skeletal stem cells (SSCs), recycling of osteoclasts via osteomorphs, and different sources of osteoblasts in development and growth have been identified. In this review, we will discuss recent advances in bone biology, and new findings on the genetic regulation of the osteogenic lineage.

## 2. Bone Formation in Embryogenesis

Bones originate from three identifiable lineages in early embryogenesis. The axial skeleton, including the spine and rib cage, are derived from the somites; craniofacial bones are generated from the neural crest and paraxial mesoderm; and the appendicular skeleton is derived from the lateral plate mesoderm [1,2,3]. Skeletal development begins with migration of mesenchymal cells to the sites of future bones, and skeletal elements are then formed via either intramembranous ossification or endochondral ossification, both of which begin with the mesenchymal cell condensation [4] (Figure 1A).

Upon mesenchymal condensation, the progenitor cells will differentiate into osteoblasts or chondrocytes for intramembranous and endochondral ossification, respectively (Figure 1B). Given that these cells are bipotent, they are often referred to as osteochondroprogenitor cells. A recent single cell transcriptomic study showed that the expression profile of osteochondroprogenitor cells derived from the neural crest and mesoderm are different [5]. For example, osteochondroprogenitors in limb buds, expressing *PRRX*1 and low levels of *SOX*9 that give rise to the cartilage template of long bones, are distinct to a neural crest-derived population identified in calvarial bones from a human embryo [5]. Further, using RNA velocity analysis for the assessment of cellular state hierarchy or lineage, osteochondroprogenitors from the limb bud can be specified into either RUNX2+ osteogenic or SOX9+ chondrogenic lineages [5].

### 2.1. Neural Crest-Derived Osteochondroprogenitors

Flat bones in the skull, the mandible, and the clavicles are examples of bone formed from neural crest-derived mesenchymal cells. Under intramembraneous ossification, they form compact or spongy bone from a layer of mesenchymal condensate. Neural crest cells are multipotent progenitors localized to the edge of the developing neural tube, expressing a specific set of transcription factors including ZIC, TFAP2, MSX1/2, SOX9/10, SNAIL1/2, PAX3/7, and MYC [6]. They undergo epithelial to mesenchymal transition (EMT), forming migratory mesenchymal cells that migrate to diverse locations in the body, including the anlagen of skull bones. MSX1/2 appear to be critical for the lineage of neural crest cells to skeletal cells, as inactivation of *Msx1* and/or *Msx2* leads to frontal bone defects with varying severity [7]. Interaction of MSX2 with TWIST is essential for proliferation and differentiation of these neural crest-derived mesenchymal cells in the formation of the skull bones [8]. Further, homeodomain-containing transcription factors, DLX2/3/5/6, are also involved in this transition from neural crest-derived cells to osteoblasts [9]. Recently, JAG1 has been shown to stimulate neural crest cell-derived osteoblast commitment [10]. JAG1, via activation of JAK2, increases STAT5 phosphorylation, promoting the osteoblast lineage through expression of *Runx2* and *Bmp2* [11]. RUNX2 is a key determining transcription factor that directs the commitment of mesenchymal progenitors towards the osteogenic lineage [12]. 

### 2.2. Mesoderm-Derived Osteochondroprogenitors 

Axial and appendicular bones are formed via endochondral ossification. In this process, a cartilage template (anlagen) of the future bone is generated containing chondrocytes, and cells surrounding chondrocytes form the perichondrium, defining the border of the template [13]. In this ossification process, chondrocytes undergo hypertrophy forming hypertrophic cartilage that is mineralized and then converted to bone. This occurs in the primary and secondary ossification sites of developing long bones, and in the cartilage growth plates located at the ends of long bones. Once formed, they are responsible for the linear growth of long bones, where there is an active proliferation of chondrocytes prior to hypertrophy. 

SOX9 is the master transcription factor to specify undifferentiated mesenchymal cells into osteochondroprogenitors and chondrocyte differentiation [14,15]. Heterozygous mutations of SOX9 result in campomelic dysplasia, severe skeletal malformation syndrome and sex reversal [16,17,18,19]. Shortly after mesenchymal condensation, SOX9, together with SOX5 and SOX6, direct expression of cartilaginous genes, including specific matrix genes such as *Col2a1* and *Acan* (Aggrecan), and facilitate chondrocyte proliferation [20]. Although expression of *Sox9* is necessary to establish the mesenchymal condensation, commitment of osteochondroprogenitors towards osteogenic lineage requires downregulation of *Sox9* and upregulation of *Runx2* [21]. SOX9 antagonizes the activities of RUNX2 and β-catenin via direct interaction [22,23]. Furthermore, inactivation of *Sox9* in differentiated chondrocytes results in a cell fate switch to an osteogenic lineage, supporting the inhibitory role of SOX9 in osteoblast differentiation [24].

### 2.3. Endochondral Ossification—Cartilage to Bone Conversion 

Chondrocyte hypertrophy represents the initiation of endochondral ossification, a process regulated by an upregulation of *Runx2* and *Col10a1* expression, concomitant with a downregulation of *Atf4* [25,26,27]. The function of collagen type X (*Col10a1*) in hypertrophic cartilage is not clear, but a role in mineralization is proposed [28]. Perichondrial cells adjacent to hypertrophic chondrocytes differentiate into osteoblasts and secrete extracellular matrix forming the periosteal bone collar [29]. The primary ossification center is then formed inside the hypertrophic region, which is the first mineralization zone of the cartilage anlagen. 

At the chondro-osseous junction, hypertrophic chondrocytes direct an invasion of blood vessels via the production of vascular endothelial growth factor (VEGF) [30]. The infiltrating blood vessels facilitate recruitment of osteoclasts to degrade the mineralized cartilage, and osteoprogenitors to initiate osteogenesis, forming the primary spongiosa with the establishment of growth plate cartilage at the epiphyseal ends of long bones [31,32] (Figure 1C). In the growth plate, a proliferative zone precedes hypertrophy. Here, downregulation of *Atf4* with hypertrophy is critical in postnatal development of the mouse, as ectopic activation of *Atf4* in hypertrophic chondrocytes leads to defective endochondral ossification, with reactivated expression of *Sox9* and pre-hypertrophic chondrocyte marker genes, such as *Col2a1*, *Ppr*, and *Ihh,* and only a low level of *Col10a1* in hypertrophy chondrocytes [26]. 

### 2.4. Transition of Hypertrophic Chondrocytes to Osteoblasts 

Although apoptosis and replacement by bone have been widely considered as the terminal fate of hypertrophic chondrocytes, recent studies using different lineage tracing mouse models (*Col2a1*-CreERT, *Sox9*-CreERT, *Acan*-CreERT, *Osx*-CreERT, *Col10a1*-Cre, and *Col10a1*-CreERT) have indicated that hypertrophic chondrocytes retain the plasticity of multi-lineage potential to differentiate into osteoblasts, adipocytes, pericytes, and stromal cells [33,34,35,36,37,38,39,40]. The hypertrophic chondrocyte-derived osteoblasts play essential roles in maintaining bone homeostasis and fracture healing [35,39].

These data support that osteoblasts can be formed in two ways, either directly differentiated from osteochondroprogenitors or via chondrogenesis prior to the osteoblast state. However, some questions remain unresolved. Chondrocyte differentiation is controlled by SOX9 at the beginning but shifts to RUNX2 during hypertrophy, therefore RUNX2 plays a dual role in chondrogenesis and osteoblastogenesis [41]. This raises the question of whether the shift in transcription regulation from SOX9 to RUNX2 primes the cells to become osteoblasts, and hypertrophic chondrocytes act as a transition cell state between chondrocyte and osteoblast.

### 2.5. Progenitor Cells for the Continued Growth of Long Bones

In addition to the proliferative and hypertrophic zones, there is a reserve zone containing mostly non-dividing chondrocytes that serve to provide a source of cells to the proliferative zone. Recent studies suggested that the growth plate is an important niche housing skeletal stem cells (SSCs). These SSCs are multipotent that can give rise to chondrocytes, osteoblasts, and adipocytes under specific differentiation conditions [42]. In humans, SSCs have been isolated from the growth plate, exhibiting specific cell surface markers (PDPN^+^CD146^-^CD73^+^CD164^+^) [43]. In mice, cell lineage studies using *Grem1*-CreERT, *PTHrP*-CreERT, and *Hoxa11*-CreERT^2^ mouse lines have demonstrated that the *Grem1^+^-, PTHrP^+^-, and Hoxa11^+^-*expressing cells behave as progenitors, giving rise to cells along the skeletal lineages [38,44,45]. Further, telomerase (*Tert*)-expressing SSCs are enriched at the time of active adolescent bone growth [46], whereas “embryonic skeletal stem/progenitor cells” (eSSPCs), marked by expression of *FOXP1/2*, have been identified in the perichondrial regions and primary ossification centers of human embryonic long bones, and a similar population is present in E15.5 mouse long bones [5]. Thus, there appears to be multiple sources of SSCs with slightly different characteristics. Why there are so many sources of SSCs or whether they can perform similar functions in bone growth, maintenance, and repair remains to be addressed. It is also possible that each subtype may have a preferential function in bone biology yet to be elucidated.

### 2.6. Appositional Bone Growth 

Radial bone growth is achieved by bone modeling during the period of active bone growth. Formation modeling on periosteal surfaces and resorption modeling on the endocortical surface lead to increases in bone width over time. Hormonal control and mechanical loading play major roles in this process [47]. Appositional bone growth is the increase in the thickness of bones by the addition of bone tissue from the marrow (endosteal) or periosteal surface of bones. This thickening and increase in bone mass continue in postnatal life, reaching a peak in humans around the age of 20–30 years old [48]. Therefore, osteoprogenitors or SSCs are expected to contribute to this growth as well as repair and remodeling processes. Osteoblasts are the cells responsible for bone matrix deposition and calcification. These cells produce extracellular matrix components such as collagen type I and osteopontin, and alkaline phosphatase that aids the mineralization process and the deposition of hydroxyapatite crystals during bone formation. 

WNT/β-catenin signaling plays a crucial role in the maturation of osteoblasts [49]. Removal of β-catenin from early osteoblastic precursors results in an arrest of osteoblast differentiation [49,50]. Recent studies have provided in vivo evidence that WNT signaling determines the osteoblastic commitment from mesenchymal progenitors, as well as the transition of hypertrophic chondrocytes to osteoblasts [51,52]. 

With appositional growth, a fraction of the mature osteoblasts become quiescent cells lining the bone surface, while active osteoblasts become embedded into the calcified bone matrix and differentiate into osteocytes as part of the osteocytic canalicular network (Figure 1D). Osteocytes are considered as terminally differentiated cells which maintain skeletal homeostasis. These cells reside in lacunae and communicate with each other through dendritic canaliculi to regulate calcium and phosphate homeostasis [53]. Osteocytes produce and transport sclerostin (*SOST*) via the canaliculi to the osteoblasts at the bone surface, as a negative regulator of bone formation through binding to LRP5/6 coreceptors, inhibiting WNT signaling [54]. Dysregulation of bone formation and maintenance results in a wide range of common diseases such as osteoporosis and numerous rare diseases of bone, such as osteogenesis imperfecta.

## 3. Transcriptional Regulation in Osteogenesis

Building functional bone during development and maintenance of bone homeostasis in adults relies on spatiotemporal activation of osteogenic transcription factors. Their expression is regulated at multiple levels, with fine-tuning requiring interactions with partners functioning to activate or suppress specific gene expression, as well as epigenetics, hormonal controls, and environmental cues (Figure 2). Dysregulation of these transcription factors and cofactors may lead to bone deformities and bone mass disorders. On the flip side, mechanisms for modulation of these factors may provide solutions for treating bone diseases.

### 3.1. Runx2 and Its Regulation

RUNX2 (runt-related transcription factor 2) belongs to the RUNT transcription factor family which is expressed in the late stage of mesenchymal condensation in skeletal development, and in osteochondral progenitor cells, acting as a “master” regulator of osteogenesis [55]. Heterozygous mutations of *RUNX2* result in cleidocranial dysplasia (CCD), an autosomal dominant dysmorphology characterized by hypoplastic clavicle, dental abnormality, and delayed bone development [56]. Its expression is critical for the osteoblastic differentiation of perichondrium progenitors and bone collar maturation [57,58]. *Runx2*-deficient mice die at birth due to respiratory failure, and bone formation is severely impaired [12].

A key function for RUNX2 is the transactivation of major bone matrix protein genes via the osteoblast-specific cis-acting element (OSE) in osteogenesis and bone formation [59,60]. Core binding factor β (CBFβ) is a co-transcription factor for RUNX2. It forms heterodimers with RUNX to regulate the expression of osteoblast genes including *Col1a1*, *Spp1* (osteopontin), *Bglap/Ocn* (osteocalcin), and *Ibsp* (bone sialoprotein) [61,62]. Numerous additional studies with chromatin immunoprecipitation (ChIP) sequencing have broadened the spectrum of osteoblast-specific genes regulated by RUNX2 [63,64]. 

The expression of *Runx2* peaks in preosteoblasts/immature osteoblasts and decreases in mature osteoblasts, indicating its essential role in early differentiation. Interestingly, overexpression of *Runx2* in osteoblasts driven by a *Col1a1* promoter results in impaired matrix production and osteopenia with multiple fractures in mice [65]. In these transgenic mice, the number of mature osteoblasts and osteocytes was greatly diminished, indicating that RUNX2 can exert a negative effect on osteoblast maturation [65]. In the divergent lineage of osteochondroprogenitors, downregulation or suppression of *Runx2/RUNX2* expression is required for chondrogenesis, however, when chondrocyte differentiation progresses to hypertrophy, *Runx2* is upregulated again [41]. The “second wave” of *Runx2/RUNX2* expression may be involved in the transition of hypertrophic chondrocytes to osteoblasts, and further studies may be required to understand this [41]. 

The transcriptional activity of RUNX2 is also modulated by other interacting partners, including HES1, SMAD, YAP/TAZ, and HAND2 [66,67,68,69]. RUNX2 is needed in regulating energy supply during bone formation. As glucose is the main nutrient during osteoblast differentiation, RUNX2 regulates the expression of *Glut1* as the major glucose transporter. The cooperative crosstalk between RUNX2 accumulation and glucose uptake favors osteoblast differentiation and whole-body glucose homeostasis [70].

TWIST1 and TWIST2 transcription factors are suppressors of RUNX2, determining the onset of osteoblast differentiation [71]. RUNX2-mediated osteoblastic gene expression only occurs when expression of *Twist1/2* decreases [71]. *Twist1* heterozygous deletion and null mutations in mice and humans, respectively, exhibit premature osteoblast differentiation in skull bones. *Twist1* heterozygosity rescues skull abnormalities in *Runx2^+/-^* mice and restores osteoblast differentiation. The interaction between TWIST and RUNX2 reduces the binding efficiency of RUNX2 to target genes. Several homeodomain-containing transcription factors, such as MSX2, DLX3, and DLX5, are also involved in regulating the level of *Runx2* expression [72]. Thus, *Msx2*-deficient mice exhibit a significant reduction in *Runx2* and *Ocn* expression, and consequently defective bones [73]. On the other hand, HOXA2 is a negative regulator of *Runx2* expression. Ectopic bone formation and *Runx2* expression are detected in the developing branchial area in *Hoxa2-*deficient mice, indicating an inhibitory role on *Runx2* expression. Finally, SATB2 represses *Hoxa2* expression via binding to its enhancer region. SATB2 synergistically promotes RUNX2-mediated *Osx* expression, but is also capable of activating *Osx* expression independent of RUNX2 [74]. 

### 3.2. Osx/Sp7 as a Downstream Target of RUNX2

Osterix (OSX), also known as SP7, is an osteoblast-specific transcription factor belonging to the Krüppel-like family [75]. It has roles in the latter stages of osteogenesis and maturation, controlling maturation to functional osteoblasts and further differentiation to osteocytes. Deletion of *Osx* in mice leads to neonatal lethality due to the failure of general bone formation, severe rib cage malformation, and a lack of expression of osteoblast genes such as *Sparc* and *Spp1* [75,76]. Its role in osteoblasts is supported by a conditional deletion in these cells using *Col1a1*-Cre (2.3 kb), giving rise to similar bone abnormalities with delayed osteoblast maturation [77]. Postnatal inactivation of *Osx* results in abnormal cartilage accumulation, the absence of trabecular bone, and impaired osteocyte maturation [78].

*Osx* is considered as a downstream target of RUNX2 as *Osx* is not expressed in *Runx2*-deficient osteoblasts, whereas expression of *Runx2* is observed in *Osx*-deficient osteoblasts [60,75,79]. Thus, during the osteogenic lineage specification, RUNX2 promotes the differentiation of mesenchymal progenitors, initiating osteogenesis, and OSX supports the maturation of functional osteoblasts. Interestingly, treatment of *RUNX2*-deficient cells with BMP2 can maintain the expression level of *OSX* via DLX5 even in the absence of RUNX2, suggesting the presence of an alternative regulatory mechanism independent of RUNX2 [80]. The transcriptional activity of OSX is also modulated by interacting partners, microRNA, and downstream targets. For example, NFATc1 was shown to control bone formation by interacting with OSX, regulating its transcriptional activity [81]. The essential role of OSX is further attributed to its regulation of osteoblast markers such as *Dkk1*, an important antagonist of WNT/β-catenin signaling [82]. Thus, an integral relationship exists between RUNX2 and OSX in complex communications among various regulators in bone cells in bone development and homeostasis.

### 3.3. Other Transcription Factors Regulating Osteoblast Differentiation

Foxhead box class O family member proteins (FoxOs) have diverse functions to promote osteogenesis at multiple steps in the osteoblast differentiation process. FOXO1 can be converted from an activator to a promoter-specific repressor of peroxisome proliferator-activated receptor γ (PPARγ) needed for adipogenesis, thus favoring osteoblastogenesis at the stage of osteoblast lineage commitment and differentiation [83]. Overexpression of *FoxO1* can significantly increase the expression of osteogenic genes such as *Runx2, Alp,* and *Ocn* in mouse mesenchymal stem cells [84]. Increased binding of FOXO1 to the promoter of *Run*x2 leads to elevated *Runx2* expression in MC3T3E1 preosteoblastic cells, indicating a direct transcriptional control in osteogenesis [85]. FOXO1 also regulates osteoblast proliferation and protein synthesis through physical interaction with ATF4, which regulates amino acid uptake and protein synthesis [86]. Consistently, conditional inactivation of *FoxO1* showed low bone mass in mice, with a reduced number of osteoblasts and synthesis of bone matrix proteins such as collagen type I [86]. 

ATF4 is a crucial regulator in bone formation, determining the onset and terminal differentiation of osteoblasts. ATF4 binds to osteoblast-specific elements, OSE1 or OSE2, in the promoter region of *OCN*, transactivating its expression. Binding to OSE2 requires RUNX2 [87], and SATB2, a nuclear matrix protein that physically interacts and stabilizes the synergistic activities of ATF4 and RUNX2 [88]. ATF4 also transactivates other osteogenic genes such as *BSP* and *OSX* [89,90].

Phosphorylation of ATF4 by RSK2 is needed for osteoblast differentiation. A lack of phosphorylation of ATF4 and hence activation of ATF4 target genes were observed in osteoblasts derived from *Rsk2-*deficient mice. Both *Rsk2-* and *Atf4-*deficient mice exhibited similar phenotypes in skeletogenesis, suggesting a genetic relationship [89]. *Atf4*-deficient mice showed delayed bone formation in embryonic development and low bone mass in adult life [89]. Phosphorylation of ATF4 by PKA, independent of RSK2, is enhanced in mice lacking neurofibromin in osteoblasts (*Nf1*_ob_), leading to increased ATF4-dependent collagen synthesis and bone formation. ATF4 is required for amino acid import in osteoblasts and is required for collagen type I synthesis [91]. Thus, a low-protein diet further reduces bone synthesis and bone mass in *Nf1*_ob_-deficient mice, whereas a high-protein diet can rescue the skeletal phenotypes in *Atf4*-deficient mice. These data support a role of ATF4 in linking nutrient and skeletogenesis.

Iroquois genes (IRX) comprise a conserved family of TALE class homeodomain-containing transcription factors. IRX proteins are involved in limb patterning and bone development. The fused toe (Ft) mouse with a 1.6 Mb deletion that included the entire Iroquois B gene cluster containing *Irx3, Irx5*, and *Irx6* displayed severe distal truncations of limbs [92,93]. In humans, homozygous missense mutations in *IRX5* give rise to Hamamy syndrome, a recessive congenital skeletal disorder [94,95,96]. Patients with a 3.2 Mb deletion at 16q12.2–13, which includes *IRX3, IRX5*, and *IRX6*, display craniofacial abnormality [96,97]. In mice, *Irx3* and *Irx5* compound deletion mutants die at mid-gestation due to cardiac defects and skeletal malformation [98,99], whereas conditional deletion of *Irx3/5* using osteoblast-specific *Osx*-Cre in mice results in osteopenia in postnatal growth [100]. IRX3 and IRX5 regulate skeletal development in a dosage-dependent manner, with bone loss severity correlating with the number of functional alleles of *Irx3/5* [40].

Bone marrow mesenchymal stem cells can give rise to chondrocytes, osteoblasts, and adipocytes. In humans, the rate of bone formation is negatively correlated with bone marrow adiposity and the adipocyte proportion is significantly increased in patients with osteoporosis [101]. Loss of WNT signaling changes the fate commitment of osteoblasts to adipocytes from mesenchymal progenitors [51]. Notably, IRX3 and IRX5 act as downstream mediators of WNT signaling to determine the fate of progenitors, fine-tuning the divergence between osteogenesis and adipogenesis [40]. Clearly, the transcriptional network determining osteogenesis is complex, and further advances in molecular and informatic technologies will play an important role to further our understanding.

### 3.4. Epigenetic Control of Osteoblast Differentiation

Epigenetics is a reversible mechanism regulating gene expression without changing the genomic sequence. The recognition, accessibility, and binding efficiency of transcription factors to cis-acting elements can be modulated by modifications of DNA or histone proteins. The role of microRNAs (miRNAs) is also considered as part of the epigenetic control of gene expression. Deficiency of *DICER*, a ribonuclease that processes miRNA precursors to mature miRNA, affects osteoblast differentiation, indicating a role for miRNA in osteogenesis [102,103,104], as recently reviewed [105]. Indeed, there are numerous studies on the epigenetic regulation of key transcription factors in osteogenesis with a strong emphasis on RUNX2 and its regulators.

#### 3.4.1. miRNA in Osteoblast Differentiation

miRNAs are small endogenous noncoding RNA molecules of around 20–22 nucleotides in length, accounting for only 1–5% of our genome, that modulate around 60% of human protein-coding genes through binding with mRNA at the 3′ untranslated regions (UTRs) [106]. miRNA that regulates osteogenesis can be classified into osteo-suppressing and osteo-enhancing types. Osteo-suppressing miRNA binds to *RUNX2* or its positive regulators, reducing their level for activation. Conversely, osteo-enhancing miRNAs function to remove repression of RUNX2. A panel of “osteo-miRNAs” (miR-23a, miR-30c, miR-34c, miR-133a, miR-135a, miR-137, miR-204, miR-205, miR-217, and miR-338) with inhibitory effects have been identified and directly target *RUNX2* [105]. Others include miR-455-3p, miR-155, and miR-6797-5p that also act on the level of *Runx2* mRNA [107,108,109], or positive regulators of RUNX2 that include BMPs, FGFs, histone acetyltransferases (HATs), CREB-binding protein (CBP), and monocytic leukemia zinc finger protein (MOZ) [110]. Activation of BMP signaling via receptors BMPR1b and BMPR2 facilitates osteogenic differentiation, activating expression of *Runx2* [111,112]. Levels of *Bmpr2* mRNA can be reduced by miR-153 and miR-100, repressing osteogenic differentiation [112,113]. *Hoxa10* and *Dlx5* are downstream targets of BMP signaling, promoting osteogenesis [114,115]. The positive effect of HOXA10 on osteogenesis is suppressed by miR-320a [116], whereas the *Dlx5* mRNA level can be modified by miR-141 and miR-200a [117]. Repressors of *Runx2* expression include SNAIL1, TWIST, and histone deacetylases (HDACs) 3–6 [118]. miR-3960 and miR-2861, clustered at the same loci, are inducible by BMP2. They assist in sustaining high levels of RUNX2 by targeting *Hdac5* and *Hoxa2*, reducing the mRNA level of these repressors. Thus, overexpression miR-3960 promotes osteoblastogenesis, while inhibition of this miRNA attenuates osteoblast differentiation [119]. miR-29b is interesting as it has multiple roles in osteogenesis [120]. It promotes osteogenesis by suppressing negative regulators such as HDAC4, TGFβ3, ACVR2A, CTNNBIP1, and DUSP2. In another study, the miR-145 level was found to be inversely correlated with the expression of osteogenic factors *RUNX2, OSX, FOXO1*, and *CTNNB1* (β-catenin) [121], and overexpression of miR-145 in C2C12 cells inhibited osteogenic differentiation, reducing the *OSX* level [122]. These findings support a role for miRNA in fine-tuning osteogenesis at all levels in the process. More recent findings on this topic can be found in a recent review [105].

Mature miRNAs can function within the cell and can be transferred from cellular cytoplasm through extracellular vesicles as circulating RNA in the systemic circulation. Circulating microRNAs have been identified in different biofluids, such as amniotic fluid, plasma, saliva, tears, and urine. Accumulating evidence shows that circulating miRNAs can be used as biomarkers for diseases [123,124]. Studies in patients with bone disorders showed that the expression profiles of circulating miRNAs between cohorts with osteoporosis bone metastases and bone fractures are different to normal individuals. miR-29b was found to be downregulated in patients with osteoporosis, while miR-124, miR-125b, and miR-148a were upregulated [125]. The functions of these circulating miRNAs in bone homeostasis are not fully understood, but the prospect of circulating miRNAs as biomarkers is an attractive option, and recently reviewed [126].

#### 3.4.2. DNA and Histone Modifications in Osteoblast Differentiation

DNA methylation is a process by which methyl groups are added to the DNA, changing the activity of a DNA segment without changing the sequence. Methylation is associated with repression of gene transcription. DNA methyltransferases (DNMTs) are the major enzymes involved in DNA methylation. Silencing of *Osx* is mediated by DNMT1/3a, whereas its activation is mediated by the DNA demethylation actions of SWI/SNF- and TET1/TET2-containing complexes. TET1 and TET2 belong to the ten–eleven translocation (*Tet*) gene family of DNA demethylases. Mice with combined *Tet1* and *Tet2* deficiency exhibited impairment in osteogenic differentiation and osteopenia, due to reduced demethylation of the *P2rX7* promoter and thus an accumulation of RUNX2-suppressing miRNAs [127].

Histone modification is an epigenetic mechanism for gene expression control. It is a post-translational modification (methylation, phosphorylation, acetylation, ubiquitylation, and sumoylation) to histone proteins, altering the architecture of chromatin. *Ocn* expression in osteoblasts is positively correlated with H4 acetylation, whereas reduced acetylation of H3 results in an inactivation of *Ocn* expression. Expression of *Hoxa10* has a positive effect on the transcription of *Runx2, Alp*, and *Ocn* [115], through chromatin hyperacetylation and trimethyl histone K4 (H3K4) methylation of these genes. Outside the genome, sirtuin 1 (SIRT1), a histone deacetylase (HDAC) that removes acetyl groups from β-catenin, is involved in osteogenesis, modulating the activity of WNT signaling [128]. 

#### 3.4.3. Modulating Epigenetic Regulators as Therapy for Bone Disorders

Targeting epigenetic regulators to modify expression control of osteogenic genes could be a novel approach to enhance bone formation for treating bone mass disorders. “Epidrugs” such as DNMT and HDAC inhibitors have been approved by the FDA or are under clinical trials to treat various diseases such as cancers and metabolic and cardiovascular disorders [129]. 5-Aza-2′-deoxycytidine functions as a DNMT inhibitor and is a lineage determinant between adipogenesis and osteoblastogenesis [130]. MSCs treated with this drug promote osteoblast differentiation via demethylation, enhancing expression of *Wnt10a, Alp, Osx, Twist1,* and *Dlx5* [131]. Valproic acid (VPA) is a short-chain branched fatty acid and is an HDAC inhibitor used in treating epilepsy. Human adipose tissue-derived stromal cells (ADSCs) treated with VPA can increased the expressions of *OSX, OPN, RUNX2,* and *BMP2*, favoring osteogenesis [132]. Sodium butyrate (NaBu), also an HDAC inhibitor, enhances osteogenesis by ERK-dependent *Runx2* activation [133], or by altering the balance between the recruitment of acetylated histone H3K9 and methylated histone H3K9 onto the *Runx2* promoter, increasing its expression level [134]. Thus, the therapeutic potential of epigenetic modifiers is promising, and a clear understanding of their roles in transcriptional control of osteogenesis will enable the development of effective epidrugs to treat bone mass disorders. 

### 3.5. Regulation of Osteoblast Survival and Death

A balanced rate of osteoblast proliferation, differentiation, and apoptosis is needed, maintaining a proper pool of osteoblasts for bone remodeling and repair at any given time. Hormonal changes with age are highly correlated with osteoblast survival. For example, gluococorticoid-induced osteoporosis is most common. Administration of prednisolone to mice can lead to increased osteoblast apoptosis. An in vitro study showed that excess glucocorticoid induces expression of pro-apoptotic factors, *Bim* and *Bak*, and decreases expression of pro-survival factor *Bcl-xL*.

Parathyroid hormone (PTH) is classically considered as a bone catabolic agent, but can elicit anabolic effects if administered correctly for treating osteoporosis [135,136]. Intermittent administration of PTH attenuates osteoblast apoptosis and thereby stimulates bone formation. PTH also increases proteasomal proteolysis of RUNX2, while inhibiting RUNX2 degradation by E3 ligase, or overexpression of *Runx2* can extend the anti-apoptotic effect of PTH [137]. Further, RUNX2 can mediate expression of survival and anti-apoptotic genes such as *Bcl-2* [137], that can promote osteoblast differentiation and survival [138]. Consistently, increased apoptosis of osteoblasts was observed in *Bcl-2*-deficient mice [139]. Of interest, in vivo and in vitro studies showed accelerated differentiation of *Bcl-2*-deficient osteoblasts, where there is an activation of FoxOs through Akt inactivation [139], as well as increased expression of *p53* and its target genes [139]. p53 has a dual role in osteoblast differentiation and survival. *p53*-deficient mice showed increased bone mass and bone formation rates, likely via enhanced osteoblast proliferation and reduced apoptosis [140]. Inactivation of Akt by p53 leads to activation of FoxOs. p53 and FoxOs have similar functions in inhibiting cell cycles and inducing cell death in osteoblasts [141]. 

## 4. Environmental Cues Regulating Osteogenesis

Bone homeostasis relies on dynamic responses to the environment such as the extracellular matrix, mechanical stress, and molecular signaling from the surrounding tissues such as the perichondrium and periosteum, as recently reviewed [142,143,144,145]. Here, we will focus on the transcriptional changes in response to hormonal changes, light/dark cycle, and microenvironment changes such as microfractures. Failure to respond to these changes will lead to bone mass disorders such as osteoporosis. 

### 4.1. Hormonal Control of Osteoblast Differentiation

PTH acts through ATF4 as the anabolic action of PTH is compromised in Atf4-deficient mice [90]. Consequently, ATF4, acting as a transcription factor, binds to a specific enhancer sequence in the OSX promoter, activating expression, thus establishing a PTH–ATF4–OSX axis in promoting osteogenesis. Estrogen is a sex hormone known to regulate osteoblast differentiation and mineralization. Estrogen deficiency is associated with bone loss and osteoporosis in aged women [146]. Estrogen receptor α (ERα) functions as a nuclear receptor binding to estrogen response elements (EREs) [147]. GATA4 and ERα are both recruited to EREs near genes expressed in osteoblasts, such as *Alp* and *Runx2* [148]. Estradiol, an estrogen, induces *GATA4* expression in osteoblasts, and binding of GATA4 precedes ERα binding [148]. Further, GATA4 is needed for histone 3 lysine 4 dimethylation at EREs, indicating GATA4 as a pioneer factor for ERα [149]. These are some examples of osteoblast differentiation and function regulated by systemic factors, however, their actions on osteogenesis are likely to be further controlled via the circadian rhythm over a 24 h daily cycle.

### 4.2. Circadian Clock Regulates Osteogenesis

Most living organisms exhibit time-dependent physiological and behavioral changes across the circadian rhythm of the daily day/night cycle. Sleeping, metabolism, the immune system, and tissue repairs are controlled by this biological clock that oscillates with the expression of a set of genes in a repeating 24 h cycle [150,151]. The central clock is located in the suprachiasmatic nucleus (SCN) of the hypothalamus, receiving and conveying light/dark cycle information with periodic signals. Transferal of neural or hormonal (such as PTH and glucocorticoids) signals is then relayed to circadian oscillators located in the peripheral tissues [152,153] (Figure 3A). 

About 3–16% of genes expressed in the peripheral tissues are rhythmic [154]. This molecular clock is controlled by core clock transcription factors that exert opposing positive (CLOCK and BMAL1) and negative (period, PER1/2/3; and cryptochrome, CRY1/2) outcomes, forming a feedback loop that follows the light/dark cycle [155,156]. CLOCK and BMAL1 form a transcriptional complex that binds to the promoter region of target genes including *PER1/2/3* and *CRY1/2*. PER and CRY interact in the cytoplasm to form a repressor complex that is phosphorylated [157,158], and translocation to the nucleus exerts a negative effect on the CLOCK–BMAL1 complex [157,159]. This activation/repression feedback loop runs synergistically in a 12 h light and 12 h darkness cycle. Interestingly, the CLOCK/BMAL1 complex also binds to the promoter region and activates transcription of histone acetyltransferase P300, and P300 binds to and acetylates RUNX2, elevating its transcriptional activity [160] (Figure 3B). This osteogenic activity seems to be suppressed by the function of CRY2 [161]. 

Disruption of the circadian clock in nightshift workers and people with sleep restriction is associated with abnormal bone metabolism and osteoporosis [162], linking the circadian rhythm to bone biology. Signals from the central clock in SCN can be transferred to bone in the form of hormones. A number of hormones such as PTH, leptin, glucocorticoids, melatonin, and ghrelin linked to bone metabolism are involved in circadian entrainment [163,164]. Indeed, our PTH level exhibits rhythmic changes [165], as with the level of leptin [166]. *Per1* expression is regulated by PTH signaling, as indicated by the high *Per1* level in a transgenic mouse with constitutively active PTH receptors in osteoblasts [167]. *Per2* expression is also responsive to PTH in a time- and dosage-dependent manner [168]. Together, these studies have provided some insights into the entrainment of the peripheral tissue clock by SCN through hormonal control.

Osteoblasts are influenced by the circadian clock in tissue homeostasis and at healing sites of fractured bones. Oscillating core circadian transcription factors and their immediate downstream targets and mediators have been demonstrated in murine calvarial bone over a 24 h cycle [169]. Positive (Bmal1) and negative (Cry1/2, Per1/2) regulators were rhythmically expressed in the expected antiphase relationship [154]. Circadian oscillation was clearly illustrated in *ex vivo* cultures of long and flat bones using *Per2-*luc reporter mice [170], and a 24 h interval rhythmic expression of *Per2* was observed in the fracture healing sites in an *ex vivo* culture study of long bone [171]. Further, a list of osteogenic “rhythmic genes” with an oscillatory expression pattern has been identified with a transcriptome microarray study that included *Ocn, Runx2, Hif1a, Stat1*, and various ligands (FGFs, WNTs, BMPs) and their receptors [169]. Indeed, 24 h rhythmic expression of *Ocn* mRNA was observed using real-time bioluminescence imaging of the skull bone from a transgenic mouse with a luciferase reporter driven under the control of an osteocalcin promoter, supporting that the expression of osteogenic genes is regulated by the circadian rhythm [172].

Further evidence of circadian regulators directly controlling osteoblastic genes lies in the expression of *Pdia3* that follows *Clock*. PDIA3 is a protein disulfide isomerase that regulates bone formation via 1α,25(OH)_2_D_3_-initiated rapid membrane signaling [173]. CLOCK regulates *Pdia3* by direct binding to its E-box promoter [174], and transcription of *Pdia3* is decreased in *Clock**^Δ^**^19^* mutant mice, supporting a direct relationship. Consequently, deficiency of major clock regulators leads to abnormal bone homeostasis in Clock-deficient (*Clock**^Δ^**^19/^**^Δ^**^19^*) [174] and *Bmal1*-deficient [175] mice, both exhibiting low bone mass. While *Per1*- and *Per2*-deficient mice are normal, *Per1^-/-^;Per2^-/-^* and *Per1^-/-^;Per2^m/m^* double mutant mice showed high bone mass phenotypes, indicating that PER1 and PER2 may compensate each other [166]. The antagonistic outcomes when disrupting positive (Clock/Bmal1) and negative (Per1/2) regulators is expected, in relation to low and high bone mass, respectively [154]. 

SIRT1 is involved in both osteogenesis and positive regulation of Clock. While crosstalk between chondrogenesis and the circadian clock through SIRT1 has been established [176], this is yet to be ascertained in osteogenesis. CLOCK mediated acetylation of BMAL1 at Lys537 through its acetylase activity, facilitating recruitment of CRY1 [177]. SIRT1 can deacetylate BMAL1 at the same site, thus releasing CRY1 suppression and prolonging CLOCK/BMAL1 positive activity [178]. SIRT1 also deacetylates PER2 to promote its degradation [179], and its inhibition leads to increased RUNX2 acetylation in mesenchymal stem cells [180]. Furthermore, SIRT1 can directly interact with RUNX2, serving as a cofactor in promoting its transcriptional activity [181]. This circadian relationship with osteogenesis and osteoblast function is exciting and will require much study to unravel this complex network.

### 4.3. Molecular Regulation at Sites of Bone Remodeling 

Bone modeling refers to the shaping of bones, usually occurring during bone formation and repair. Bone remodeling is a continuous cycle of bone renewal for tissue homeostasis. Osteoclasts are the bone-resorbing cells derived from progenitors of macrophages under the control of macrophage colony-stimulating factor (M-CSF) and receptor activator of nuclear factor-kB ligand (RANKL) [182]. In bone remodeling, osteoclasts are recruited to sites of microfractures, creating an acidic environment to dissolve minerals, and secrete digestive enzymes (cathepsins and collagenases) to remove the old bone. The fate of osteoclasts was unclear until recently, when cell-tracing and real-time imaging showed osteoclasts are recycled through fission to a distinct cell type termed “osteomorphs” [183]. Osteomorphs have a distinct molecular signature from marcophages and osteoclasts and are able to fuse again to form activated osteoclasts [183] (Figure 1D). As old bone tissue is removed, osteoblasts are mobilized to rebuild the site, depositing new bone matrix enriched in collagen type I. The newly formed bone matrix (osteoid) is then sequentially mineralized. Dysregulating the balance between bone resorption and formation will lead to bone mass disorders, the most common being osteoporosis. With aging, a depletion of progenitors for bone repair and remodeling is thought to be associated with poor bone healing and osteoporosis in the elderly. 

Communication between osteoclasts and osteoblasts is critical in the regulation of bone remodeling [184]. RANKL, a ligand produced by osteoblasts, signals to RANK, its receptor that is expressed on the surface of osteoclasts and is needed for osteoclast differentiation and function [185]. Osteoblasts also produce and secrete osteoprotegerin (OPG), a decoy receptor for RANKL, in fine-tuning bone remodeling by limiting the availability of RANKL [185]. Following menopause, a decline in estrogen leads to excessive bone modeling and imbalanced osteoclast and osteoblast activities, as estrogen serves to limit the amount of RANKL produced by osteoblasts [185]. On the other hand, osteoclasts can influence osteoblast formation and differentiation through secretion of soluble factors that promote (S1P, CTHRC1, and C3) or suppress (SEMA4D) osteoblast differentiation [186]. Additionally, as bone matrix is being degraded, entrapped signaling molecules (TGF-β and IGF-1) are released, favoring osteoblast differentiation [187,188]. In addition, direct contact between osteoclasts and osteoblasts via cell surface ligand–receptor pairings, such as EFNB2–EPHB4, FAS–FASL, and NRP1–SEMA3A pathways, regulate cell proliferation, differentiation, and survival [186].

A canalicular network connects osteocytes in mature bone, allowing physical communication between osteocytes. This network extends to osteoblasts at the bone surface, regulating bone homeostasis. The communication between osteocytes and osteoblasts is important, as conditional ablation of osteocytes in mice leads to fragile bone caused by poor bone signaling [189]. TGIF1 (TG-interacting factor 1), a homeodomain transcription factor expressed in both osteoblasts and osteocytes, has a synergistic role in regulating the expression of *Sema3e* in osteoblasts [190]. Expression of *Tgif1* in osteocytes controls the level of SOST that modulates WNT signaling in osteoblasts at the bone surface. Reduced WNT signaling in osteoblasts can lead to an enhanced expression of *Sema3e,* and the secretion of SEMA3E has a negative effect on osteoclast function [190]. *Tgif1* is also a target of PTH via AP-1, providing an important insight into the molecular and cellular relationships between the major cell types in bone.

## 5. Concluding Remarks

Rapid advances in biomedical technologies have enabled numerous new areas of bone research not possible before. We have gained tremendous insights into the many “origins” of osteoblasts during embryogenesis and in postnatal life. Depletion or “aging” of osteoprogenitor cells such as of the SSCs is likely to be a cause of age-related bone loss. Leveraging these cells for treatments is an attractive and potentially practical approach. Understanding the transcriptional and environmental cues of osteogenesis for the bone maintenance and mobilization of osteoprogenitor cells in repair is clearly important, as well as the endogenous niche for SSCs in adult tissues. Moreover, we must address the relevance and translational value of animal studies to humans in our study design.

## Figures and Tables

**Figure 1 ijms-22-05445-f001:**
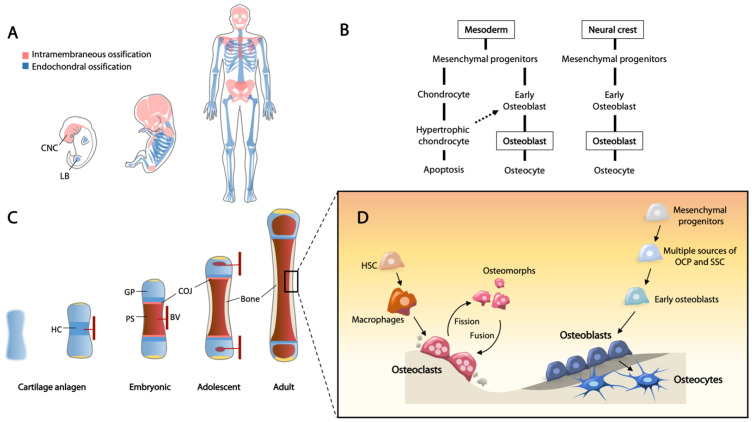
Current understanding of osteogenesis and bone remodeling. (**A**) Flat bones are formed through intramembranous ossification, cells are originated from the cranial neural crest (CNC). Limb bones are formed through endochondral ossification (details shown in (**C**)), cells are originated from the mesoderm-derived limb bud (LB) mesenchyme. (**B**) Two major routes for osteoblast differentiation. Mesoderm cells give rise to mesenchymal osteochondroprogenitors (OCPs) which can diverge into chondrocytic and osteoblastic lineages. Chondrocytes undergo hypertrophy and a portion of them differentiate into osteoblasts at the chondro-osseous junction. Neural crest-derived mesenchymal progenitors can differentiate directly to osteoblasts during intramembraneous ossification. (**C**) Endochondral ossification is a process of converting cartilage to bone and is essential for bone elongation. Cartilage anlagen of a future bone forming in the limb bud during embryogenesis. Chondrocyte hypertrophy (HC) initiates in the center of the anlagen where blood vessels (BVs) invade, bringing in osteoprogenitors and bone marrow cells. The primary spongiosa (PS) separates the cartilage into proximal and distal growth plates (GPs). From childhood to adolescence, there is an active proliferation of chondrocytes prior to hypertrophy, and the mineralizing cartilage is replaced by bone at the chondro-osseous junction (COJ). Thickening of cortical bone continues from birth to puberty when the GPs become inactive. (**D**) Bone remodeling maintains the integrity and homeostasis of bone in adulthood. Osteoclasts are bone resorptive cells originated from hematopoietic stem cells (HSCs). They remove microfractured segments of bone and mobilize osteoblasts to form new bone. Osteomorphs are a novel cell type generated through fission of osteoclasts. Subsequent fusion of osteomorphs can reform active osteoclasts. Multiple sources of skeletal stem cells (SSCs) and OCPs have been identified as the source of osteoblasts for bone formation. Some of the mature osteoblasts are embedded into the osteoid and further differentiate into osteocytes which have a critical role in bone remodeling coordination.

**Figure 2 ijms-22-05445-f002:**
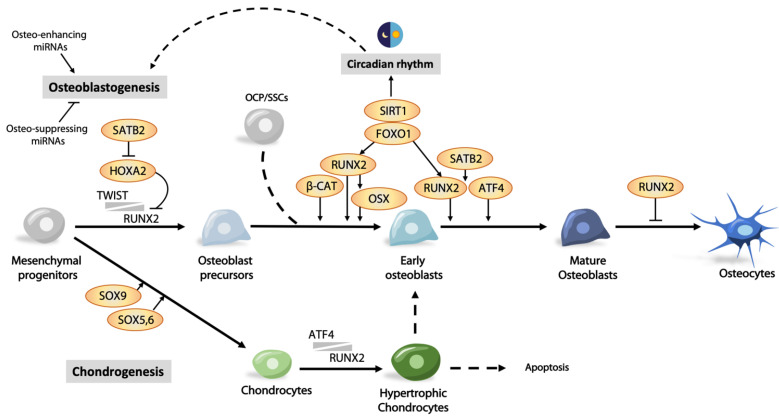
Transcriptional regulation of osteoblast differentiation. SOX9 and RUNX2 are major fate determinants of mesenchymal progenitors to chondrogenesis and osteoblastogenesis, respectively. Cells can “detour” to chondrogenesis or commit to an osteoblast lineage. RUNX2 is the master transcription factor that regulates multiple steps in osteoblast commitment and differentiation. Its transcriptional activity is controlled at multiple levels such as transcriptional co-factors, inhibitors, osteo-enhancing and -suppressing miRNAs, and environmental cues such as light–dark cycle.

**Figure 3 ijms-22-05445-f003:**
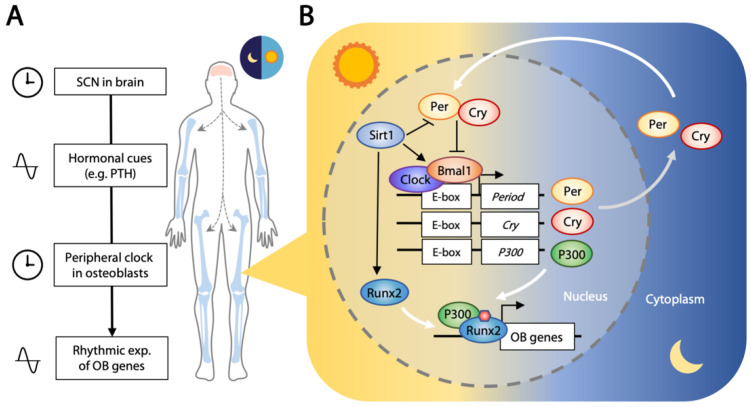
Circadian clock in bone. (**A**) Suprachiasmatic nucleus (SCN) in the hypothalamus receives the 24 h light–dark signals and conveys them in the form of nerve or hormonal signals. The rhythmic level of hormone controls the peripheral clock in bone, hence leading to rhythmic expression of osteoblastic (OB) genes. (**B**) The molecular clock involves the positive regulators CLOCK and BMAL1 which bind to the E-box elements and activate expression of circadian negative regulators PER and CRY. PER and CRY inhibit activities of CLOCK and BMAL1 to form a feedback loop that occurs within a period of 24 h. CLOCK/BMAL1 can bind to the E-box region and activate expression of P300 which subsequently promotes the acetylation of histone 3 and facilitates the formation of a transcriptional complex with RUNX2 to drive expression of osteoblastic genes. Sirt1 has dual roles in the circadian clock and osteogenesis. It binds CLOCK/BMAL1 in a circadian manner and promotes the deacetylation and degradation of PER and is a positive regulator of RUNX2.

## Data Availability

Not applicable.

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
