# Peer review of "Regulation and Role of Transcription Factors in Osteogenesis"

_ijms, 2021, doi:10.3390/ijms22115445_

Round 1

Reviewer 1 Report

The review Regulation and role of transcription factors in osteogenesis written by Chan et al.  aims to summarise molecular signalisation in the crucial cell types of bone with focus on their commitment, differentiation and functions. The senior author of the group previously published several papers in bone research and thus seems to be oriented in the field. The review is clearly organised, written in good English with minimum of typing errors. The work is based on selection of solid original papers including recently published research. In the review, three figures are included to summarise bone development and differentiation of essential bone cells, the most important transcriptional factors involved in differentiation of osteoblasts and chondroblasts, and circadian regulation in bone. One of the major advantages of this work is the concept of bone as a complex tissue. Further, I appreciate the paragraph with epigenetic and circadian regulation.

My specific comments are listed below:

Molecular signals and transcription factors are described in detail for osteoblasts, however, osteocytes and osteoclasts are not so deeply described.

P1: Could authors refer to osteocytes and osteoclasts in Abstract?

P2 L54-56: Is the composition of the sentence correct?

P2 L58: „osteochondroprogenitors from the limb bud can be specified into either RUNX2+“  instead of „osteochondroprogenitors from the limb bud can specified into either RUNX2+“

P2 L74: „They remove microfractured segments of bone“ instead of „They remove old bones with microfractures“

P3 L81: Please make the complete list of intramembranous bone or add „for instance“ to the sentence.

P3 L92: „transition“ instead of „lineage“

P3 L106: Please briefly specify interstitial and lateral growth.

P3 L123: None of references supports Atf4 information.

P3 L121-127: First, the bone collar appears and hypertrophy of chondrocytes is a consequence.

P4 L133-4: „Here, downregulation of Atf4 with hypertrophy is critical in postnatal development of mouse“ instead of „Here, downregulation of Atf4 with hypertrophy is critical“

P4 L161: “PDPN+CD146- CD73+CD164+“ instead of  “PDPN+CD146- CD73+CD146+“

P7 L246-248: repeating information

P7 L270: „but is also capable“ instead of „but also capable“

P8 L331: Please specify that the loss of WNT determines commitment of osteoblasts to adipocytes.

P10 L404: “Epidrugs” has as DNMT- and HDAC-inhibitors have been approved“ a typing error?

P10 L430: Please indicate that PTH is classically considered to be a bone catabolic agent.

P11 L486: Please refer to glucocorticoids in general, or specify the type.

Fig. 1 A: The intramembranous bones are not only present in calvarial bone. The Figure should reflect complete distribution of intramembranous bones in human body.

Fig. 1C: The growth plate (epiphyseal cartilage) is reduced in adults, and thus the scheme of adult bone should be modified to better.

Please consider following topics to be included:

Molecular signals in perichondrium/periosteum

Cell death-related pathways in osteoblasts

Role of ECM in osteogenesis

Reviewer 2 Report

This review described the main regulatory roles of transcription factors and their role on osteogenesis. The authors describe the basic molecular events that control the generation of bone-forming cells, emphasising the regulation of bone development and homeostasis by transcription factors. Interestingly, they also refer to important regulatory mechanisms that have introduced a new era in this field, such as epigenetic regulation and circadian clocks and used up-to-date citations, e.g. osteomorphs. The text has a very good flow and is well-written and well-structured. The authors are sufficiently concise and explicit.

Minor comments:

1) Please use one type of abbreviation for CreERT transgenic mice. In the present form, it is a mix of capitals, upper case letters etc. CreERT or CreERT2.

2) Indeed, ATF4 is a crucial regulator of bone formation. However, essential citations are missing. For example, PMID: 17141628.

3) Please use appropriate capitalization of gene and protein symbols. For humans, gene symbol: RUNX2, protein symbol: Runx2 and for rodents, gene symbol: Runx2, protein symbol: RUNX2.

4) The section describing the rope of microRNAs on osteogenesis can be enriched since now numerous miRs are known to have multiple roles in the process. For example, the circulating levels of certain miRs have been found altered in osteoporotic patients. Importantly, an illustration for this section would be of great value for the reader, like Fig3 for circadian regulation.

Reviewer 3 Report

Interesting review.  I have no hesitation in recommending it for publication.

Author Response

Thank you very much for your comments.

Reviewer 4 Report

The manuscript of Chan et al. is a very good review about the various transcription factors involved in osteogenesis, providing detailed information of the pathways and factors that control osteoblast differentiation. It is a complex, well-written, and up-to-date review about osteogenesis and bone remodeling, current topics of biological and clinical significance.

However, there are some points that the authors need to address, and minor errors in the manuscript that need to be corrected.

- A list of abbreviations would facilitate the reading of the text.

- Lines 68, 69, and 100. It should be “anlagen” and not “analgen”.

- Line 137. The phrase “bone collar that progresses to the formation of the growth plate” is misleading and must be rewritten.

- Line 161: There is a mistake in the manuscript text about the cell markers of the SSCs described in reference [42]. As shown in that reference, the correct phenotype is “PDPN+CD146-CD73+CD164+” (the last one CD is CD164, and not CD146, as shown in the manuscript text).

- Line 179. It should be “matrix components such as Collagen” and not “matrix such as Collagen”.

- Line 404. It should be “”Epidrugs” as” and not “”Epidrugs” has as”.

- Line 404. It should be “or are under” and not “or under”.

- Line 545. “As old bone tissue is removed” is more appropriate than “As old bones are removed”.

- Line 545. It should be “to rebuild“ and not “to the rebuild“.
